# Deep-Graph-Sprints: Accelerated Representation Learning in Continuous-Time Dynamic Graphs

**Ahmad Naser eddin**                                                                    *ahmad.eddin@feedzai.com*
*Feedzai, Portugal*
*Departamento de Ciência de Computadores, Faculdade de Ciências, Universidade do Porto, Portugal*

**Jacopo Bono**                                                                          *jacopo.bono@feedzai.com*
*Feedzai, Portugal*

**David Aparício**                                                                       *daparicio@dcc.fc.up.pt*
*Departamento de Ciência de Computadores, Faculdade de Ciências, Universidade do Porto, Portugal*

**Hugo Ferreira**                                                                        *hugo.ferreira@feedzai.com*
*Feedzai, Portugal*

**Pedro Ribeiro**                                                                        *pribeiro@dcc.fc.up.pt*
*Departamento de Ciência de Computadores, Faculdade de Ciências, Universidade do Porto, Portugal*

**Pedro Bizarro**                                                                        *pedro.bizarro@feedzai.com*
*Feedzai, Portugal*

**Reviewed on OpenReview:** *https://openreview.net/forum?id=0uwe0z2Hqm*

## Abstract

Continuous-time dynamic graphs (**CTDGs**) are essential for modeling interconnected, evolving systems. Traditional methods for extracting knowledge from these graphs often depend on feature engineering or deep learning. Feature engineering is limited by the manual and time-intensive nature of crafting features, while deep learning approaches suffer from high inference latency, making them impractical for real-time applications. This paper introduces Deep-Graph-Sprints (**DGS**), a novel deep learning architecture designed for efficient representation learning on CTDGs with low-latency inference requirements. We benchmark DGS against state-of-the-art (SOTA) feature engineering and graph neural network methods using five diverse datasets. The results indicate that DGS achieves competitive performance while inference speed improves between 4x and 12x compared to other deep learning approaches on our benchmark datasets. Our method effectively bridges the gap between deep representation learning and low-latency application requirements for CTDGs.

## 1 Introduction

Graphs serve as a foundational structure for modeling and analyzing interconnected systems, with applications spanning in computer science, mathematics, and life sciences. Recent studies have emphasized the critical role of dynamic graphs, which capture evolving relationships in systems like social networks and financial markets (Costa et al., 2007; Zhang et al., 2020; Zhou et al., 2020; Majeed & Rauf, 2020; Febrinanto et al., 2023).

Graph structure representation is crucial for encoding complex graph information into low-dimensional embeddings that are usable by machine learning models. This task is particularly challenging for dynamic graphs. Traditional graph feature engineering methods rely on manually crafted heuristics to capture graph characteristics, necessitating domain knowledge and considerable time to engineer and test new

features (Bilot et al., 2023). In contrast, graph representation learning, especially through graph neural networks (**GNNs**), automates this process by learning compact embeddings of graph structures (Hamilton et al., 2017b). Despite growing interest in this field, research has predominantly focused on static graphs, overlooking the dynamic nature of many real-world systems (Perozzi et al., 2014; Grover & Leskovec, 2016; Hamilton et al., 2017a).

Dynamic graphs are categorized into Discrete Time Dynamic Graphs (**DTDGs**) and Continuous Time Dynamic Graphs (**CTDGs**) (Rossi et al., 2020). DTDGs are viewed as a sequence of snapshots at set intervals, while CTDGs are seen as a continuous stream of events, such as adding a new edge, which updates the graph's structure with each occurrence. This paper aims to advance the state-of-the-art (SOTA) in representation learning for CTDGs.

Existing methods for handling CTDGs (Dai et al., 2016; Kumar et al., 2019; Xu et al., 2020; Rossi et al., 2020) often face computational constraints, leading to high-latency inference, thus limiting their practicality for real-time applications. Approaches such as asynchronous operation and truncated backpropagation have been employed to mitigate these issues, but they introduce compromises in representation accuracy and the learning of long-term dependencies (Rossi et al., 2020; Wang et al., 2021).

This paper introduces a novel architecture for the representation learning of CTDGs, designed to overcome existing limitations and provide low-latency, efficient representation learning. Our approach employs forward-mode automatic differentiation, specifically real-time recurrent learning (**RTRL**) (Williams & Zipser, 1989), within a customized recurrent cell structure. This enables low-latency inference, efficient computation, and optimized memory usage, while preserving representation accuracy and the ability to capture long-term dependencies. The contributions of this work are as follows:

- We identify the limitations in current methodologies for graph representation learning, highlighting their computational inefficiencies and their challenges in capturing long-term dependencies, see Sections 2,5.

- We introduce Deep-Graph-Sprints (**DGS**), a method for real-time representation learning of CTDGs, optimizing latency and enhancing the ability to capture long-term dependencies, see Sections 3.

- We benchmark DGS against SOTA methods, in node classification and link prediction tasks, demonstrating on par predictive performance while achieving significantly faster inference speed—ups up to 12x faster than TGN-attn (Rossi et al., 2020), and up to 8x faster than both TGN-ID (Rossi et al., 2020) and Jodie (Kumar et al., 2019). Detailed results are provided in Section 4.

## 2 Background: Overview of Automatic Differentiation Modes

Deep learning depends significantly on credit assignment, a process identifying the impact of past actions on learning signals (Minsky, 1961; Sutton, 1984). This process is essential for reinforcing successful behaviors and reducing unsuccessful ones.

The capability of assigning credit in deep learning models depends on the differentiability of learning signals enabling the use of Jacobians for this purpose (Cooijmans & Martens, 2019). A key technique in this context is automatic differentiation (**AD**), a computational mechanism for the derivation of Jacobians through a predefined set of elementary operations and the application of the chain rule, applicable even in programs with complex control flows (Baydin et al., 2018). In AD, depending on the direction of applying the chain rule, three strategies stand out: forward mode, reverse mode (often termed backpropagation), and mixed mode. Forward mode involves multiplying the Jacobians matrices from input to output. Reverse mode, a two-phase process, first executes the function to populate intermediate variables and map dependencies, then calculates Jacobians in reverse order from outputs to inputs (Baydin et al., 2018). Mixed mode combines these approaches.

Temporal models, such as recurrent neural networks (**RNNs**) and GNNs for temporal graphs, pose specific challenges for backpropagation due to their memory-intensive requirements. The memory complexity for storing intermediate states across a history significantly impacts the feasibility of full backpropagation.

For instance, in an RNN with sequence length $l$ and state size $d$, backpropagation-through-time exhibits computational and memory complexities of $\mathcal{O}(l \times d^2)$, posing scalability issues for long sequences (Baydin et al., 2018). To mitigate these challenges, truncated backpropagation through time (**TBPTT**) optimizes resource usage by limiting the backpropagation horizon, thus reducing both computational and memory demands. However, TBPTT's constraint on the temporal horizon restricts its ability to capture long-term dependencies, impacting model performance over extended sequences (Williams & Peng, 1990).

Forward-mode AD, exemplified in real-time recurrent learning (**RTRL**), offers an alternative by facilitating online updates of the parameters, which is particularly advantageous for models requiring the retention of information over extended durations (or sequence length). Despite its benefits for capturing long-term dependencies with reduced memory overhead ($\mathcal{O}(d^2)$), RTRL's computational demand ($\mathcal{O}(d^4)$) limits its practicality in large-scale networks (Williams & Zipser, 1989; Cooijmans & Martens, 2019).

In summary, while full backpropagation through time has high memory demands, TBPTT presents a compromise by reducing memory and computational needs at the expense of long-term dependency capture. Forward mode AD (RTRL in our case) addresses both long-term retention and memory efficiency but is restricted by its computational complexity.

To address these limitations, our approach leverages RTRL, thus benefiting from its low memory footprint, and combines it with a custom architecture that reduces its computational complexity to match that of backpropagation. This design effectively mitigates the computational challenges typically associated with forward-mode AD, and allows the model to capture long-term dependencies unlike TBPTT.

## 3 Method

In this section we detail our low latency node representation learning method, namely Deep-Graph-Sprints (**DGS**). We start by explaining the main components that form its architecture and then we detail each one of them. Furthermore, we detail the training paradigm that distinguishes our method, highlighting its memory demands and ability to capture long-term dependencies compared to existing approaches through its RTRL-based approach. Additionally, we explain our method during inference, demonstrating how it achieves low latencies.

### 3.1 Architecture

The DGS method is developed to handle a stream of edges. As shown in Figure 1, the system processes each incoming edge to derive a task-specific score, applicable for any ML task such as classification.

Similar to established approaches in this domain, e.g., (Rossi et al., 2020; Kumar et al., 2019), our DGS method is divided into two key components:

1. Embedding Recurrent Component (ER): dedicated to representation learning, where each node or edge in the graph is mapped from high-dimensional, complex graph structures to a lower-dimensional embedding space.

2. Neural Network (NN): responsible for decision-making processes, such as classification. It uses the embedding provided by the ER component to generate a task specific output.

The ER component is particularly noteworthy for its role in updating the embeddings of nodes or edges, thereby enriching them with detailed attributes and relationships context within the network. These embeddings are then input into the neural network, which is tailored to specific applications. For instance, in node classification, the network evaluates each node associated with a new edge, with the score reflecting the network's interpretation from the representations provided by the ER component.

### 3.1.1 Embedding Recurrent Component (ER):

This subsection delineates the ER component, the core mechanism within our methodology that processes dynamic graph data. Building upon *Graph-Sprints* (Eddin et al., 2023), a low-latency graph feature engi-

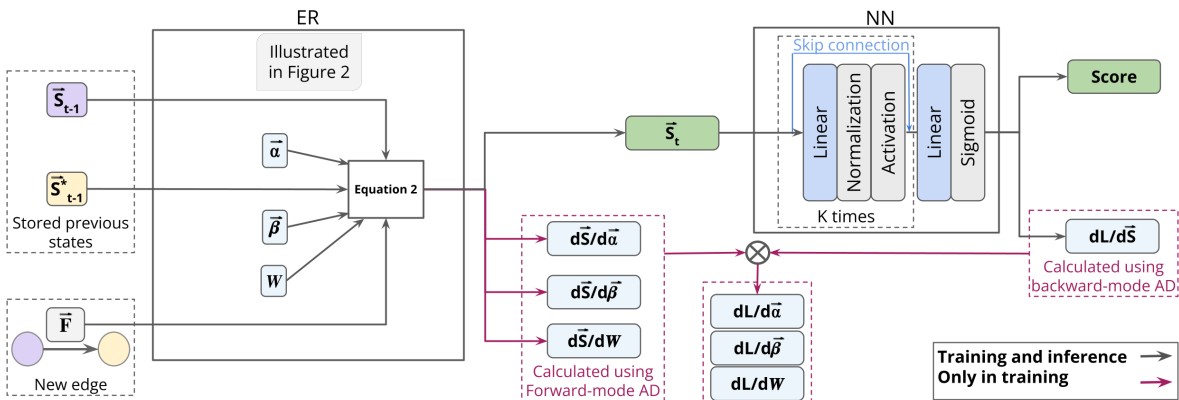

Figure 1: Schematic representation of the DGS architecture. The diagram illustrates the workflow from receiving new edge, through the generation of embeddings for nodes or edges, to the application of a neural network to generate a task specific score. Furthermore, the diagram elucidates the computation of gradients through the application of mixed-mode AD. Equation 2, and Figure 2 provide more details about the ER component.

neering approach, that is formalized in Equation 1. A node's embedding is determined by integrating its historical state, the historical state of its immediate neighbor (with which it shares the current edge), and the attributes of the current edge.

$$\vec{S}_t = \beta \vec{S}_{t-1} + (1 - \beta)\left((1 - \alpha)\vec{\delta}(\vec{F}_t) + \alpha \vec{S}_{t-1}^*\right) \tag{1}$$

Here, $\vec{S}_t$ denotes the state of a node at time $t$, evolving from its previous state $\vec{S}_{t-1}$ and the state of its immediate neighbor $\vec{S}_{t-1}^*$, while also incorporating the current edge's features $\vec{F}_t$, encoded through the $\vec{\delta}$ function, which bucketizes these features. The $\vec{\delta}$ function maps numerical features into a one-hot encoded vector based on predetermined buckets (e.g., buckets corresponding to specific percentiles). This bucketization provides a deterministic representation of the input features, which requires manual tuning to define the buckets and results in sparse, high-dimensional feature vectors. The coefficients $\alpha$ and $\beta$ are scalar forgetting factors that modulate the impact of neighborhood and past information on the current state.

Although *Graph-Sprints* demonstrates rapid processing capabilities and performs on par with leading techniques, it faces several limitations in practical applications. These include the complex and time-consuming tuning processes required for its feature extraction and decision-making components. For instance, to tune the forgetting coefficients or to define the bucketed features edges of the $\vec{\delta}$ function for every feature. Moreover, the model's expressivity is constrained by the uniform application of scalar forgetting coefficients across all features, limiting its ability to capture the unique temporal dynamics of each feature. Finally, the use of large number of buckets per feature significantly increases the memory requirements especially in datasets with many features. In contrast, our proposed methodology, while drawing inspiration from Graph-Sprints, goes beyond traditional feature engineering by employing a dynamic learning mechanism for embeddings. The state update equation in our methodology is illustrated in Equation 2:

$$\vec{S}_t = \vec{\beta} \odot \vec{S}_{t-1} + (1 - \vec{\beta}) \odot \left((1 - \vec{\alpha}) \odot \left(\bigg\|_{i=1}^{m} \vec{\sigma}(W_i \vec{F}_t)\right) + \vec{\alpha} \odot \vec{S}_{t-1}^*\right) \tag{2}$$

In this equation, we introduce vectorized forgetting coefficients, $\vec{\alpha}$ and $\vec{\beta}$, each corresponding in dimensionality to the state vector, and modulate the weighting between current and historical information, and between self and neighborhood information, respectively. Each dimension of $\vec{\alpha}$ and $\vec{\beta}$ corresponds to unique forgetting rates for each embedded feature in the state vector, which itself consists of a vector as we will detail below.

The embedding matrix $W$ is tasked with mapping input features into a vector of the same size as the state vector.

Inspired by the Graph-Sprints feature representations, where the buckets that belong to the same feature sum to one, we utilize the softmax function ($\vec{\sigma}$) to achieve analogous representations. Moreover, we employ softmax temperature scaling (Guo et al., 2017) where higher temperatures result in softer distributions. DGS enhances model expressiveness and optimizes memory usage by incorporating multiple softmax functions, each applied to a segment of the product between the embedding matrix $W$ and the feature vector $\vec{F_t}$. The notation $\|_{i=1}^{m}$ denotes the concatenation of the results obtained by applying the $m$ softmax functions. Each function is applied to the product of the $i$-th portion of the embedding matrix, $W_i$, and the features values $\vec{F_t}$, as illustrated in Figure 2. This strategy not only aids in reducing computational and memory demands by limiting the dependency of each state element to a specific segment of the embedding matrix and thereby lowering the Jacobian's dimensionality but also introduces a modular structure.

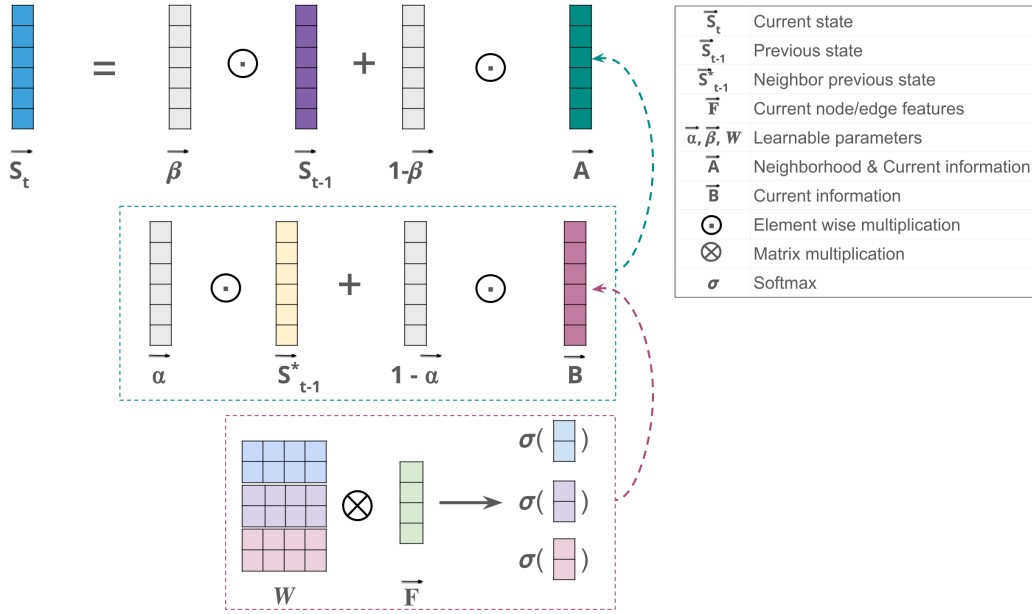

Figure 2: Schematic illustration of state calculation based on Equation 2. This example demonstrates the computation of node state at time $t$ with a state size of $s = 6$, three softmaxes ($m = 3$), and thus two rows per softmax from the embedding matrix $W$ ($h = s/m = 2$). The number of input features is $f = 4$.

### 3.1.2 Neural Network (NN)

The NN component is a feedforward neural network, that encompasses multiple layers. The configuration of this component is subject to optimization depending on the task at hand. This optimization includes decisions such as the number of layers, the size of each layer, and the incorporation of normalization layers.

Although Equation 1 (GS) and Equation 2 (DGS) present similarities. There are several aspects that distinguish *Graph-Sprints* and DGS methods. GS is a feature engineering approach with fixed embeddings, while DGS employs deep learning with learnable embeddings. Moreover, parameter optimization in GS is separate, whereas DGS allows end-to-end optimization. Table 4 compares both methods.

### 3.2 Training Process

The design of DGS methodically incorporates forward-mode AD for learning the ER component, whereas, the subsequent NN component, processing the embeddings generated by the ER component, utilizes reverse-mode AD. This hybrid approach effectively leverages the strengths of both paradigms, namely, learning

long-term dependencies (as detailed in Section 2), and ensuring efficient learning while accommodating the memory constraints and structural complexities of graph data.

In typical ML scenarios, the complexity of forward-mode AD limits its applicability. Nonetheless, forward-mode AD is applicable in situations requiring a manageable number of Jacobian computations, offering efficient Jacobian propagation through computational graphs. In the DGS method, the feasibility of forward-mode AD is supported by two main factors. First, DGS is dominated by elementwise multiplications, where different elements of a state vector are not mixed together. Second, the implementation of multiple softmax functions limits the dependency of each state element to a segment of the embedding matrix $W$, thus reducing the computational and memory requirements.

The element-wise multiplication between the state vectors $\vec{S}$ and the parameter vectors $\vec{\alpha}$ and $\vec{\beta}$ optimizes the calculation of Jacobians $\frac{\partial \vec{S}}{\partial \vec{\alpha}}$ and $\frac{\partial \vec{S}}{\partial \vec{\beta}}$, achieving a computational and memory complexity of $\mathcal{O}(s)$, where $s$ represents the size of the state vector. In contrast, performing these operations via matrix multiplication, implying $\alpha$ and $\beta$ are $(s \times s)$ matrices, would increase the complexity to $\mathcal{O}(s^3)$. Regarding the embedding matrix $W$, with dimensions $s \times f$ (embedding size by the number of features), the application of a single softmax function over the entire embedded vector would result in Jacobians $\frac{\partial \vec{S}}{\partial W}$ with dimensions $f \times s^2$, leading to computational and memory complexities of $\mathcal{O}(f \times s^2)$. However, DGS mitigates this through the deployment of multiple softmax functions, each managing a segment of the state. With $m$ softmax functions, each addressing a subset of $h = s/m$ rows, the computational and memory requirements are effectively reduced to $\mathcal{O}(f \times h \times s)$, demonstrating the method's efficiency in optimizing both computational and memory resources. Furthermore, one can easily fix $h$ to a predetermined value and optimize the state size $s$ to be multiples of this parameter. Therefore, assuming a fixed $h$, the total computational complexity scales linearly with respect to the state size $s$. This property demonstrates a better scaling than backpropagation, which scales with $s^2$. These factors collectively justify the selection of forward-mode AD for the differentiation process in the ER component of our architecture. The Jacobians updates were implemented manually using PyTorch (Fey & Lenssen, 2019).

In the NN component, number of learnable parameters varies based on model architecture, primarily involving the network's weights. The parameters of the NN component are optimized using backpropagation, and to implement that we also leverage the functionalities of PyTorch. As a result, the architecture of the DGS method employs a mixed-mode AD approach, as illustrated in Figure 3.

Compared to the GS approach, allowing the key parameters ($\vec{\alpha}$, $\vec{\beta}$, and $W$) to be learned from the data overcomes the limitations of separate tuning processes, thereby simplifying the training procedure. Additionally, this enables the use of vectorized forgetting coefficients instead of scalar ones, which significantly enhances the model's expressivity.

### Mini-Batch Training

To expedite the training process, we employ mini-batch training, wherein the input comprises a batch of edges. In cases where a singular node appears multiple times within a single batch, each occurrence is associated with the same prior node state, which represents the most recent state prior to the batch's execution. This methodology implies that nodes contained within the same batch do not utilize the most current information due to the prohibition of intra-batch informational exchange. One can also implement a batch strategy similar to the one implemented by the Jodie method (Kumar et al., 2019), where batch size is dynamic and nodes only appear once in the same batch.

### 3.3 Inference Process

The inference phase is characterized by the absence of gradient computation, which simplifies the overall procedure. In the streaming context, as elaborated in Section 3.1, the occurrence of an edge triggers an update in the states of the nodes interconnected by this edge, employing Equation 2 for the update mechanism. Subsequently, these updated states are ingested by the NN component. The nature of the input to the NN component is task-dependent: it may constitute a singular node state for node classification tasks, or the concatenation of two node states for tasks such as link prediction.

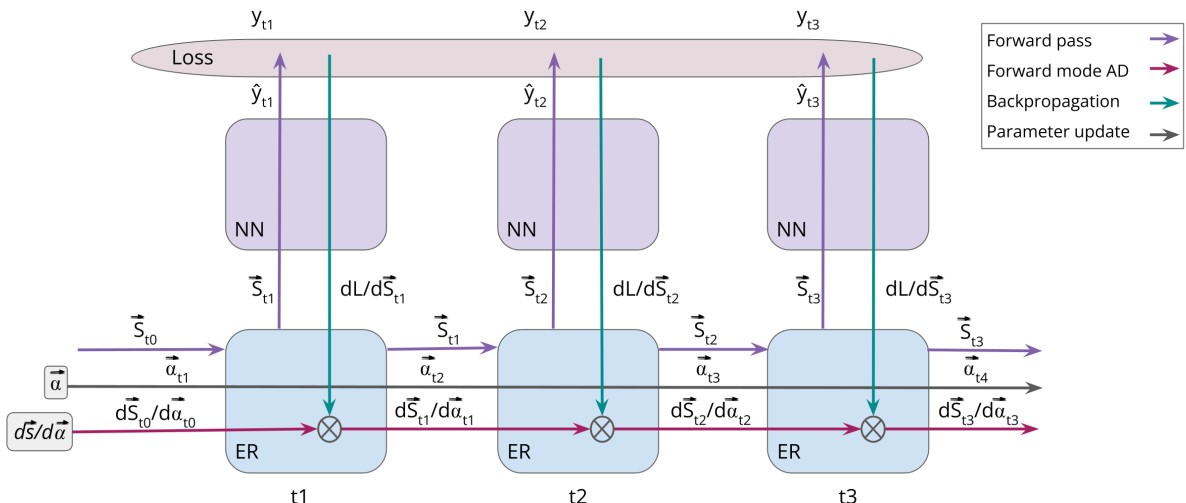

Figure 3: Recurrent Training Process: This figure illustrates the steps involved over three successive timesteps, focusing on the derivative calculation for a single learnable parameter $\vec{\alpha}$ using a mixed-mode approach. It combines forward mode differentiation for the ER component with backpropagation for the neural network classifier. This methodology extends to update other parameters (i.e., $\vec{\beta}$, $W$). This hybrid approach enables an efficient solution that effectively captures long-term dependencies.

**Mini-Batch Inference**

To accelerate inference in scenarios suitable for batch processing, we employ a mini-batch inference strategy. This approach updates the states of nodes or edges within each batch simultaneously. When a node appears multiple times within the same batch, as in the training phase, each instance is linked to the same prior node state, which is the most recent state before the batch's execution. Consequently, there is no exchange of states within the batch. Note that this is optional and similarly to mini-batch procedure in training we can leverage a different strategy.

Following the parallel updates, the aggregated states are inputted into the neural network (NN) component. This step generates a batched output tailored to the task, whether it involves node classification, link prediction, or any other relevant activity.

It is important to note that the DGS method is fully online and supports both single-sample and mini-batch inference, offering flexibility depending on the scenario. Mini-batch inference, when suitable, improves both training and inference speeds even further. In our experiments (Section **??**), we utilize mini-batch inference to ensure comparability with other experimental setups used in state-of-the-art studies.

# 4 Experiments and Results

## 4.1 Experimental Setup

The efficacy of our methodology was evaluated through the node classification and link prediction tasks across five different datasets. This include three open-source external datasets and two proprietary datasets from the anti-money laundering (**AML**) domain.

**Baselines**

We compare the performance of our method against several baselines. The first simple baseline, called Raw, trains a machine learning model using only raw edge features. Another baseline is Graph-Sprints (Eddin et al., 2023) a graph feature engineering method, which we refer to by GS. The GS baseline uses the same ML classifier used by the Raw baseline but diverges in the features used for training— GS employs Graph-Sprints encoded features, whereas Raw employs the raw edge features.

An additional baseline set comprises SOTA GNN methods, specifically TGN (Rossi et al., 2020). Our TGN implementation, based on the default PyTorch Geometric implementation, differs from the original paper by restricting within-batch neighbor sampling, for a more realistic scenario.

For the node classification tasks, our TGN implementation diverges slightly from the default PyTorch Geometric implementation, which was originally implemented for link prediction, by updating the state of target nodes with current edge features *before* classification. In contrast, for link prediction, this update occurs *after* the classification decision, aligning with the PyTorch Geometric implementation. These settings are typical for node classification and link prediction, respectively, and both GS and DGS follow the same setup[1].

Several TGN variants were used: TGN-attn, aligning with the original paper's best variant, TGN-ID, a simplified version focusing solely on memory module embeddings, and Jodie, which utilizes a time projection embedding with gated recurrent units. TGN-ID and Jodie baselines, which do not necessitate neighbor sampling, were chosen for their lower-latency attributes compared to TGN-attn. All GNN baselines (TGN-ID, TGN-attn, and Jodie) used a node embedding size of 100.

**Optimization**

The hyperparameter optimization process utilizes Optuna (Akiba et al., 2019) for training 100 models. Initial 70 trials are conducted through random sampling, followed by the application of the TPE sampler. Each model incorporated an early stopping mechanism, triggered after 10 epochs without improvement. Table 6 enumerates the hyperparameters and their respective ranges employed in the tuning process of DGS and the baselines.

Importantly, the state size for DGS is fixed to 100 in the node classification task, achieved by setting the product of the number of softmax functions and the number of rows per softmax to 100 ($m \times h = 100$). This aligns with the configurations of other GNN baseline models (TGN-ID, TGN-attn, and Jodie) to ensure comparability. In the link prediction task, we set the DGS state size to 250 because a state size of 100 was insufficient for achieving comparable performance. Despite this larger state size compared to the GNN baselines, the DGS method has, on average, 2.5 times fewer learnable parameters than the TGN-attn baseline. Additionally, only 35% of the learnable parameters in $DGS$ on average are attributed to the $ER$ component, with the remaining 65% belonging to the classification head (further detail in Table 9).

**Datasets**

We leverage five different datasets, all CTDGs and labeled. Each dataset is split into train, validation, and test sets respecting time (i.e., all events in the train are older than the events in validation, and all events in validation are older than the events in the test set). Three of these datasets are public (Kumar et al., 2019) from the social and education domains. In these three datasets, we adopt the identical data partitioning strategy employed by the baseline methods we compare against, which also utilized these datasets. The other two datasets are real-world banking datasets from the AML domain. Due to privacy concerns, we can not disclose the identity of the FIs nor provide exact details regarding the node features. We refer to the datasets as FI-A and FI-B. The graphs in this use case are constructed by considering the accounts as nodes and the money transfers between accounts as edges. Table 5 shows the details of all the used datasets.

---

[1]In the original GS paper, link prediction for the GS was performed using the same setup as node classification, i.e. updating the state before classification. We believe it is more fair to use the link prediction setup as all other models, hence our GS results on link prediction tasks are not directly comparable to the original paper. Similarly, the TGN baselines in node classification tasks used the link prediction setup in the original paper, hence those results are also not directly comparable here.

## 4.2 Node classification

In the node classification task, given the dataset characteristics detailed in Table 5, we address binary classification with class imbalance. To evaluate performance, we calculate the area under the ROC curve (ROC-AUC), referred to as AUC for brevity, by plotting the True Positive Rate (TPR) against the False Positive Rate (FPR) across various classification thresholds, and then computing the area under the curve. The AUC is calculated using the 'sklearn' library in python.

The results for **node classification** are detailed in Table 1, displaying the average test AUC $\pm$ std for the external datasets and the $\Delta$ AUC for the AML datasets. To obtain these figures, we retrained the best model identified through hyperparameter optimization across 10 different random seeds.

It is important to note that the GNNs and GS baselines leverage the latest edge information, similar to the DGS method. This means they update the node state with the most recent information before classifying the node.

We have highlighted the **best** and second-best performing models for each dataset. To provide an overview, we include a column showing the average rank, representing the mean ranking computed from all datasets. DGS achieves either the highest or the second-highest scores in four out of the five datasets. The exception is the Mooc dataset, where GNN baselines surpass our method. We did note that there is some overfitting of the GNN baselines. This is due to the extreme scarcity in positive labels, which resulted in the validation metrics being badly correlated with the test metrics for these baselines.

Counter-intuitively, although not reported here[2], we observed that the performance of the GNN baselines improved when evaluated without leveraging the latest edge features, which could indicate a reduced overfitting to the validation dataset.

Table 1: Node classification results using public and internal datasets.

| Method | AUC $\pm$ std | | | $\Delta$AUC $\pm$ std | | Average rank |
|---|---|---|---|---|---|---|
| | Wikipedia | Mooc | Reddit | FI-A | FI-B | |
| Raw | 58.5 $\pm$ 2.2 | 62.8 $\pm$ 0.9 | 55.3 $\pm$ 0.8 | 0 | 0 | 6 |
| TGN-ID | 69.3 $\pm$ 0.5 | **86.3** $\pm$ 0.8 | 56.2 $\pm$ 3.7 | +1.2 $\pm$ 0.1 | +24.3 $\pm$ 1.8 | 3.4 |
| Jodie | 68.8 $\pm$ 1.3 | 86.1 $\pm$ 0.4 | 56.2 $\pm$ 2.1 | +1.4 $\pm$ 0.1 | +25.0 $\pm$ 0.6 | 3.2 |
| TGN-attn | 70.5 $\pm$ 4.1 | 86.0 $\pm$ 0.9 | 55.6 $\pm$ 6.1 | +0.9 $\pm$ 0.2 | +22.5 $\pm$ 2.5 | 4.2 |
| GS | **90.7** $\pm$ 0.3 | 75.0 $\pm$ 0.2 | **68.5** $\pm$ 1.0 | +1.8 $\pm$ 0.5 | **+27.8** $\pm$ 0.4 | 2 |
| DGS | 89.2 $\pm$ 2.2 | 78.7 $\pm$ 0.6 | 68.0 $\pm$ 1.9 | **+3.6** $\pm$ 0.2 | +26.9 $\pm$ 0.3 | 2.2 |

## 4.3 Link Prediction

For the link prediction task, the evaluation process generates $n - 1$ negative edges for each positive edge, where $n$ denotes the number of nodes (possible destinations) in the graph. We then measure the mean reciprocal rank (**MRR**), which indicates the average rank of the positive edge. An MRR of 50% implies that the correct edge was ranked second, while an MRR of 25% implies it was ranked third. Additionally, we measure Recall@10, which represents the percentage of actual positive edges ranked in the top 10 scores for every edge.

We retrain the hyperparameter-optimized model using 10 random seeds and report the average test MRR $\pm$ standard deviation and Recall@10 $\pm$ standard deviation in Table 2. Evaluations were conducted in both transductive (T) and inductive (I) settings. The transductive setting involves predicting future links of nodes that could be observed during training, while the inductive setting involves predictions for nodes not encountered during training.

We identified the **best** and second-best models. DGS demonstrated competitive performance in link prediction. It outperformed the GNN models by approximately 10% in MRR on the Mooc dataset and showed

---

[2]We believe it is more fair to compare the performance using the same setup. For the interested reader, the results when updating the state after the classification are reported in the GS paper Eddin et al. (2023).

improved performance on the Reddit dataset. However, it underperformed compared to the baselines on the Wikipedia dataset. To provide a comprehensive overview, we included a column in Table 2 that displays the average rank, representing the mean ranking derived from all datasets, calculated using MRR. Notably, our DGS model achieved the highest average performance in both transductive and inductive settings.

Table 2: DGS: Link prediction results using public datasets.

| | Method | Wikipedia | | Mooc | | Reddit | | Average rank |
| --- | --- | --- | --- | --- | --- | --- | --- | --- |
| | | MRR | Recall@10 | MRR | Recall@10 | MRR | Recall@10 | |
| T | TGN-ID | $46.6 _{\pm 2.4}$ | $67.3 _{\pm 2.1}$ | $15.3 _{\pm 6.6}$ | $36.9 _{\pm 18.0}$ | $41.3 _{\pm 4.2}$ | $57.4 _{\pm 3.5}$ | 4.3 |
| | Jodie | $\underline{65.3} _{\pm 1.3}$ | $\underline{78.4} _{\pm 0.2}$ | $15.3 _{\pm 3.9}$ | $38.1 _{\pm 11.8}$ | $42.4 _{\pm 4.3}$ | $59.9 _{\pm 2.9}$ | 2.7 |
| | TGN-attn | $\mathbf{66.7} _{\pm 1.3}$ | $\mathbf{78.3} _{\pm 0.6}$ | $\underline{16.9} _{\pm 3.6}$ | $\underline{42.1} _{\pm 11.2}$ | $40.0 _{\pm 9.2}$ | $56.8 _{\pm 8.9}$ | 2.7 |
| | GS | $54.7 _{\pm 1.1}$ | $64.4 _{\pm 0.9}$ | $4.0 _{\pm 0.4}$ | $5.1 _{\pm 0.5}$ | $\mathbf{55.5} _{\pm 1.2}$ | $\mathbf{65.2} _{\pm 11}$ | 2.7 |
| | DGS | $53.9 _{\pm 1.3}$ | $63.9 _{\pm 0.6}$ | $\mathbf{25.6} _{\pm 4.0}$ | $\mathbf{49.0} _{\pm 5.3}$ | $\underline{51.0} _{\pm 0.9}$ | $\underline{64.8} _{\pm 0.3}$ | $\mathbf{2.3}$ |
| I | TGN-ID | $\underline{62.3} _{\pm 1.2}$ | $\underline{75.1} _{\pm 0.5}$ | $13.8 _{\pm 5.9}$ | $31.1 _{\pm 15.8}$ | $41.6 _{\pm 2.5}$ | $59.6 _{\pm 1.9}$ | 2.7 |
| | Jodie | $57.9 _{\pm 0.7}$ | $73.1 _{\pm 0.6}$ | $16.7 _{\pm 2.6}$ | $41.2 _{\pm 6.4}$ | $37.9 _{\pm 4.2}$ | $57.0 _{\pm 3.1}$ | 4.0 |
| | TGN-attn | $\mathbf{65.6} _{\pm 2.4}$ | $\mathbf{75.8} _{\pm 0.7}$ | $\underline{17.7} _{\pm 2.0}$ | $\underline{42.1} _{\pm 5.2}$ | $48.1 _{\pm 2.2}$ | $\underline{64.7} _{\pm 0.9}$ | 2.0 |
| | GS | $55.0 _{\pm 2.1}$ | $62.8 _{\pm 1.2}$ | $2.8 _{\pm 0.2}$ | $3.6 _{\pm 0.3}$ | $\underline{49.4} _{\pm 1.1}$ | $59.5 _{\pm 1.3}$ | 4.0 |
| | DGS | $59.3 _{\pm 2.5}$ | $68.5 _{\pm 2.4}$ | $\mathbf{26.0} _{\pm 3.9}$ | $\mathbf{48.2} _{\pm 3.3}$ | $\mathbf{56.9} _{\pm 30.9}$ | $\mathbf{68.5} _{\pm 22.5}$ | $\mathbf{1.7}$ |

## 4.4 Inference Runtime

DGS has a primary goal of achieving reduced inference times. Comparative latency assessments were conducted amongst DGS, Graph-Sprints, and baseline GNN models. These assessments involved processing 200 batches, each containing 200 events, across distinct datasets (Wikipedia, Mooc, and Reddit) for node classification task. The average times were computed over 10 iterations. Tests were performed on a Linux PC equipped with 24 Intel Xeon CPU cores (3.70GHz) and an NVIDIA GeForce RTX 2080 Ti GPU (11GB). Note that all experiments, including those for link prediction and node classification mentioned in the previous sections, used the same machine.

As depicted in Figure 4, DGS exhibited significant speed advantages. On the Reddit dataset, it was more than ten times faster than the TGN-attn GNN baseline. For the smaller datasets, this speed enhancement ranged approximately between 5 and 6 times, while maintaining a competitive speed with the low latency GS baseline. Notably, the runtime of DGS remained stable and was not influenced by the number of edges in the graph, as demonstrated in Figure 4. In the Wikipedia and Reddit datasets, DGS consistently took 0.24 seconds. In contrast, TGN-attn exhibited a runtime increase from 1.2 seconds in the Wikipedia dataset to approximately 3 seconds in the Reddit dataset. This stability suggests that we may obtain higher speed gains in larger or denser graphs, especially considering that inference times in the TGN-attn baseline are impacted by the number of graph edges. Moreover, When benchmarked against other GNNs baselines (TGN-ID, Jodie), DGS consistently demonstrated significantly lower inference latency.

In comparison to the GS framework, known for its low latency, DGS generally exhibited marginally superior speed, especially noticeable in the Wikipedia and Reddit datasets, with latencies of 0.24 versus 0.29 seconds. This performance gain can be attributed to the higher feature count in these datasets (172 features), which potentially increases the processing time for GS due to the elevated feature volume. In contrast, for the Mooc dataset, which has only 7 edge features, GS showed a slight gain in speed (0.24 versus 0.28 seconds).

## 4.5 Ablation Study

In this section we conduct a comparative analysis of three distinct variants of the DGS methodology, differentiated primarily by their parameterization complexity.

The initial variant, designated as DGS-s, represents the most basic approach wherein *scalar* parameters $\alpha$ and $\beta$ are learned. Moreover, instead of employing a learnable embedding matrix, DGS-s adopts the static embedding function utilized by Graph-Sprints. The subsequent variant, DGS-v, retains the fixed embedding

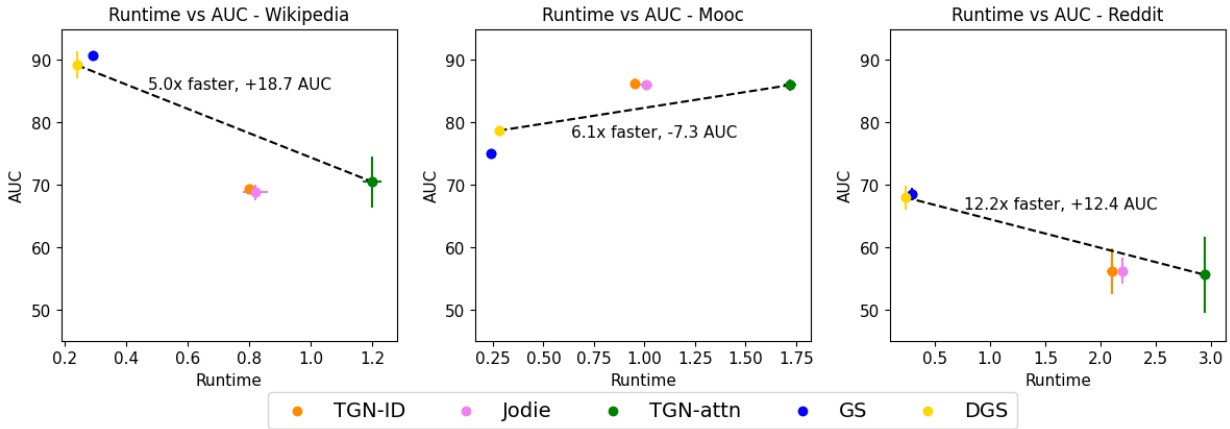

Figure 4: Trade-off between AUC and runtime.

function but transitions to *vectorized* parameters, $\vec{\alpha}$ and $\vec{\beta}$. This modification aims to explore the effects of increasing these parameters' complexity on the performance of the model. The final variant, referred to as DGS, not only incorporates *vectorized* parameters $\vec{\alpha}$ and $\vec{\beta}$ but also integrates a learnable embedding matrix. This approach aims to assess the impact of learnable feature embedding matrix.

Table 3 presents the results of the three variants for **node classification** across five different datasets. It displays the average test AUC $\pm$ standard deviation for the external datasets and the $\Delta$ AUC for the AML datasets. The DGS variant showes to be on average the best variant.

Table 3: DGS: Ablation study, Node classification results

| Method | AUC ± std | | | ΔAUC ± std | | Average rank |
|--------|-----------|---|---|------------|---|------|
| | Wikipedia | Mooc | Reddit | FI-A | FI-B | |
| DGS-s | $88.2 _{\pm 0.6}$ | $73.8 _{\pm 0.5}$ | $65.8 _{\pm 0.8}$ | $+1.8 _{\pm 0.3}$ | $25.8 _{\pm 0.7}$ | 3 |
| DGS-v | $\mathbf{91.0} _{\pm \mathbf{0.3}}$ | $75.2 _{\pm 0.3}$ | $67.2 _{\pm 0.4}$ | $+3.2 _{\pm 0.1}$ | $+26.7 _{\pm 0.2}$ | 1.8 |
| DGS | $89.2 _{\pm 2.2}$ | $\mathbf{78.7} _{\pm \mathbf{0.6}}$ | $\mathbf{68.0} _{\pm \mathbf{1.9}}$ | $\mathbf{+3.6} _{\pm \mathbf{0.2}}$ | $\mathbf{+26.9} _{\pm \mathbf{0.3}}$ | $\mathbf{1.2}$ |

Further ablation studies demonstrating the advantages of using forward-mode AD and softmax normalization are detailed in Appendix A.4. Moreover, the differences in inference speed are discussed in Appendix A.5

## 5 Related Work

Graph representation learning is essential for converting complex graph structures into embeddings usable by machine learning models. This section provides an overview of the existing algorithms. Additionally, given the focus of this paper, we explore approaches for low latency in graph representation learning.

### 5.1 Graph Representation Learning

Most existing graph representation learning methods focus on static graphs, thereby neglecting temporal dynamics (Perozzi et al., 2014; Tang et al., 2015; Grover & Leskovec, 2016; Hamilton et al., 2017a; Ying et al., 2018). Dynamic graphs, which evolve over time, introduce additional complexities. A common approach is to use DTDGs by considering the dynamic graph a series of discrete snapshots and apply static methods (Sajjad et al., 2019), but this approach fails to capture the full spectrum of temporal dynamics.

To address this limitation, more advanced techniques have been developed to better handle CTDGs. These methods include incorporating time-aware features or inductive biases into the architecture(e.g., (Nguyen et al., 2018; Jin et al., 2019; Lee et al., 2020; Rossi et al., 2020)).

For instance, methods like *DeepCoevolve*(Dai et al., 2016) and *Jodie*(Kumar et al., 2019) train two recurrent neural networks (RNNs) for bipartite graphs, one for each node type. In these models, the previous hidden state of one RNN is also used as an input to the other RNN, allowing interaction between the two and effectively performing single-hop graph aggregations.

*TGAT*(Xu et al., 2020) introduces temporal information through time encodings, enhancing the model's ability to capture dynamic changes. *TGN*(Rossi et al., 2020) extends this approach by incorporating a memory module in the form of an RNN, providing a more robust framework for handling temporal data. Further refinement is seen in Jin et al. (2020), where the discrete-time recurrent network of *TGN* is replaced with a *NeuralODE*, modeling the continuous dynamics of node embeddings for more accurate representations.

The methods described above either leverage random-walks or graph neural networks (GNNs) to extract neighborhood information and understand graph structure. Random-walk-based methods are often hindered by high computational and memory costs, as noted by Xia et al. (2019). Solutions to mitigate these challenges include techniques such as *B_LIN*(Tong et al., 2006), *METIS*(Karypis & Kumar, 1997), and *RWDISK* (Sarkar & Moore, 2010), which offer approximations of random walks.

GNNs are powerful for representation learning of graphs, but adapting them to extensive datasets poses challenges. When the graph does not fit in memory, sampling the neighbors of a node may necessitate costly disk reads. While various sampling strategies have been proposed, integrating these with temporal data is complex. Enhancing the scalability of GNNs for real-time applications remains a critical area of ongoing research (Jin et al., 2023).

### 5.2 Low-latency Graph Representation Learning

This section reviews methods for low-latency graph representation learning. For example, *APAN* (Wang et al., 2021) aims to reduce inference latency by decoupling expensive graph operations from the inference module, executing the costly *k*-hop message passing asynchronously. While *APAN* enhances inference efficiency, it may use outdated information due to its asynchronous updates, which could impact overall performance. In contrast, our method, *Deep-Graph-Sprints*, addresses latency without compromising the freshness of the information used.

Furthermore, Liu et al. (2019) present a real-time algorithm for graph streams that updates node representations based on the embeddings of 1-hop neighbors of a node of interest, and ignoring its attributes. Chu & Lin (2024) propose ETSMLP, a model that leverages an exponential smoothing technique to model long-term dependencies in sequence learning. However, ETSMLP is tailored specifically for sequence modeling and is not applicable to graph-based tasks out-of-the-box. *Graph-Sprints* (Eddin et al., 2023) offers a feature engineering approach that approximates random walks for low-latency graph feature extraction. Unlike our approach, *Graph-Sprints* requires extensive hand-crafting of features.

In addition to inference optimization, several methods address the reduction of computational costs in GNNs. *HashGNN*(Wu et al., 2021) employs MinHash to generate node embeddings suitable for link prediction tasks, grouping similar nodes based on their hashed embeddings. Another approach, *SGSketch*(Yang et al., 2022), introduces a streaming node embedding framework that gradually forgets outdated edges, leading to speed improvements. Unlike our approach, *SGSketch* primarily updates the adjacency matrix and focuses on the graph structure rather than incorporating additional node or edge attributes.

## 6 Conclusions

CTDGs are essential for representing connected and evolving systems. The computational and memory demands associated with performing lookups to sample multiple neighbors limit their feasibility in low-latency scenarios.

In this paper, we introduce the real-time graph representation learning method for CTDGs, named DGS. This novel approach addresses the latency challenges associated with current deep learning methods. It also obviates the need for manual tuning and domain-specific expertise, which are prerequisites for traditional feature extraction methods. The architecture design makes the use of real-time recurrent learning (RTRL)

feasible, which in turn can help to learn long-term dependencies and to use online learning. To validate the effectiveness and applicability of DGS, we conducted a thorough evaluation using two internal AML datasets and additional datasets from various fields, thereby demonstrating its versatility.

Future work includes exploring alternative normalization functions or activation functions beyond softmax and incorporating advanced optimization algorithms such as Adam to replace the current use of stochastic gradient descent for updating DGS parameters during forward-mode AD. Additionally, a significant enhancement under consideration is enabling input-dependent adaptability for the parameters $\vec{\alpha}$ and $\vec{\beta}$, aiming to improve the model's responsiveness to varying input features and enhance overall performance, similar to approaches used in gated recurrent neural networks.

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

# A   Appendix

## A.1   Comparing Graph-Sprints and Deep-Graph-Sprints

Deep-Graph-Sprints builds upon and addresses the limitations of Graph-Sprints, resulting in both notable similarities and key distinctions, as outlined in Table 4.

Table 4: Comparison of *Graph-Sprints* and *Deep-Graph-Sprints*

| Aspect | Graph-Sprints | Deep-Graph-Sprints |
|---|---|---|
| **Components** | Embedding + classifier | Embedding + classifier |
| **Input embeddings** | Hard-coded (feature engineered) | Learnable |
| **Optimization of embedding and classifier parameters** | Separate | End-to-end |
| **Embedding size** | Fixed and large (bucketing of raw features) | Tuneable (learnable embeddings) |
| **Embedding expressivity** | Scalar forgetting coefficients | Vector forgetting coefficients |

Table 5: Information and data partitioning strategy for public (Kumar et al., 2019), and proprietary datasets. In the public datasets, we adopt the identical data partitioning strategy employed by the baseline methods we compare against, which also utilized these datasets. In the proprietary datasets(FI-A, FI-B) due to privacy concerns we provide approximated details

|  | **Wikipedia** | **Mooc** | **Reddit** | **FI-A** | **FI-B** |
|---|---|---|---|---|---|
| #Nodes | 9,227 | 7,047 | 10,984 | $\approx$400,000 | $\approx$10,000 |
| #Edges | 157,474 | 411,749 | 672,447 | $\approx$500,000 | $\approx$2,000,000 |
| Label type | editing ban | student drop-out | posting ban | AML SAR | AML escalation |
| Positive labels | 0.14% | 0.98% | 0.05% | 2-5% | 20-40% |
| Duration | 30 days | 29 days | 30 days | $\approx$300 days | $\approx$600 days |
| Used split (%) | 70-15-15 | 60-20-20 | 70-15-15 | 60-10-30 | 60-10-30 |

## A.2   Data: Specifications and Characteristics

Table 5 outlines the specifications and key characteristics of the five datasets utilized in the experiments described in Section 4.

## A.3   Hyperparameters Ranges

Table 6 enumerates the hyperparameters and their respective ranges employed in the tuning process of *DGS* and the baselines. All represents the hyperparameters that are common to all the used methods (i.e., DGS, GS, and GNN)

Table 6: Hyperparameters ranges for **DGS** and baseline methods.

| **Method** | **Hyperparameter** | **min** | **max** |
|---|---|---|---|
| **DGS** | DGS learning rate ($\eta$) | $10^{-4}$ | $10^3$ |
| **DGS** | Number of softmaxes ($m$) | 10 | 50 |
| **DGS** | Softmax temperature ($T$) | 1 | 10 |
| **GS** | $\alpha$ | 0.1 | 1 |
| **GS** | $\beta$ | 0.1 | 1 |
| **GNN** | Memory size | 32 | 256 |
| **GNN** | Neighbors per node | 5 | 10 |
| **GNN** | Num GNN layers | 1 | 3 |
| **GNN** | Size GNN layer | 32 | 256 |
| **ALL** | Learning rate | $10^{-4}$ | $10^{-2}$ |
| **ALL** | Dropout perc | 0.1 | 0.3 |
| **ALL** | Weight decay | $10^{-9}$ | $10^{-3}$ |
| **ALL** | Num of dense layers | 1 | 3 |
| **ALL** | Size of dense layer | 32 | 256 |

## A.4   Ablation Study

We evaluate the performance of different variants of the DGS method on the node classification task. We compare three distinct approaches:

- **DGS-bp:** DGS with the typical truncated backpropagation strategy as used in SOTA methods such as TGN and JODIE.

- **DGS-sum:** DGS using a divide-by-sum normalization technique, which simplifies the normalization process by summing activations across the network. Equation 3 illustrates the process for a single partition in the $W$ matrix (analogous to a single softmax operation in our proposed approach).

$$\vec{E}_i = \frac{W_i \vec{F}_t}{\sum (W_i \vec{F}_t) + \varepsilon} \tag{3}$$

- **DGS (proposed):** Our proposed method employs forward-mode AD and utilizes softmax normalization.

As shown in Table 7, our proposed DGS method outperforms both DGS-bp and DGS-sum, demonstrating the advantage of using forward-mode AD and softmax normalization for this task.

Table 7: Node classification performance comparison across DGS variants (DGS-bp, DGS-sum, and the proposed DGS method). Detailed descriptions of these variants are provided in Appendix A.4.

| Method | AUC ± std | | |
|---|---|---|---|
| | **Wikipedia** | **Mooc** | **Reddit** |
| DGS-sum | 84.5 ± 9.1 | 71.3 ± 3.9 | 53.7 ± 7.4 |
| DGS-bp | 85.8 ± 0.4 | 72.6 ± 13.2 | 63.2 ± 0.7 |
| DGS (proposed) | **89.2** ± 2.2 | **78.7** ± 0.6 | **68.0** ± 1.9 |

## A.5 Comparison of Inference Speed

We evaluate and compare the inference speed of two different methods: DGS (proposed) and DGS-sum. The comparison is based on their performance across various datasets. Table 8 summarizes the average inference times, in seconds, along with their standard deviations. The results indicate that both methods exhibit similar inference speeds across the different datasets.

Table 8: Comparison of inference speed using different normalization functions

| Data | DGS (proposed) | DGS-sum |
|---|---|---|
| Wikipedia | 0.24 ± 0.002 | 0.25 ± 0.009 |
| Reddit | 0.24 ± 0.003 | 0.23 ± 0.0004 |
| Mooc | 0.28 ± 0.003 | 0.27 ± 0.0003 |

## A.6 Comparison of Learnable Parameter Counts

Table 9 provides a comprehensive comparison of the learnable parameter counts between DGS and TGN-attn for the link prediction task across inductive and transductive settings, underscoring differences in model complexity.

Table 9: Comparison of the number of learnable parameters between DGS and TGN-attn in the link prediction task, evaluated in both inductive (I) and transductive (T) settings. The ratio indicates the relative proportion of parameters in TGN-attn compared to DGS.

| | Mooc | | Wiki | | Reddit | |
|---|---|---|---|---|---|---|
| | I | T | I | T | I | T |
| DGS-NN component | 80,145 | 80,145 | 57,665 | 23,537 | 84,945 | 80,145 |
| DGS-ER component | 2,250 | 2,250 | 43,500 | 43,500 | 43,500 | 43,500 |
| DGS (total) | 82,395 | 82,395 | 101,165 | 67,037 | 128,445 | 123,645 |
| TGN-attn | 154,221 | 308,785 | 362,422 | 197,572 | 100,426 | 282,340 |
| Ratio | 1.9 | 3.7 | 3.6 | 2.9 | 0.8 | 2.3 |

