# OpenReview forum: "Deep-Graph-Sprints: Accelerated Representation Learning in Continuous-Time Dynamic Graphs"
_TMLR — Accepted by TMLR_

### Review · Reviewer_8VfK · 2024-08-22

**Summary Of Contributions:**

The paper presents a new deep learning architecture called Deep-Graph-Sprints (DGS) for efficient representation learning on continuous-time dynamic graphs (CTDGs). Traditional methods for extracting knowledge from CTDGs rely on either feature engineering, which is limited by manual efforts and time, or deep learning, which suffers from high inference latency, making it unsuitable for real-time applications. DGS addresses these issues by providing low-latency inference while maintaining competitive performance. The authors benchmark DGS against state-of-the-art feature engineering and graph neural network methods using five diverse datasets. The results show that DGS improves inference speed up to 12x compared to other deep learning approaches, effectively bridging the gap between deep representation learning and low-latency application requirements for CTDGs.

**Audience:**

Yes

**Claims And Evidence:**

Yes

**Requested Changes:**

Additional empirical studies are recommended, particularly since the current performance is not promising.

**Strengths And Weaknesses:**

Strengths ：

1. Efficient Low-Latency Inference: The DGS method significantly reduces inference latency, especially when dealing with large datasets, with inference speeds up to 12 times faster than other deep learning methods.
2. Automated Representation Learning: DGS avoids the manual tuning required by traditional feature engineering methods, reducing time and resource consumption through automated representation learning.
3. Extensive Experimental Validation: DGS has been extensively validated on five different datasets, including public and proprietary datasets, demonstrating its competitiveness in node classification and link prediction tasks.

Weaknesses：
1. Model Complexity: The introduction of vectorized forgetting coefficients and learnable embedding matrices in the ER component increases the complexity of the model, potentially making the training process more intricate.
2. Limitations of Mini-Batch Inference: Although mini-batch inference accelerates the inference speed, it may lack the flexibility of single-sample inference in certain scenarios.
3. Dataset Dependence: Experimental results show that DGS performs less well on some datasets, such as the Mooc dataset, possibly due to overfitting or underfitting issues related to the specific characteristics of the dataset.

---

> ### Author Response · Authors · 2024-09-18
> **Response to Reviewer 8VfK**
>
> We thank reviewer 8VfK for the valuable feedback. Below we replicate the identified weaknesses/Requested Changes and how we addressed these concerns:
>
> > Model Complexity: The introduction of vectorized forgetting coefficients and learnable embedding matrices in the ER component increases the complexity of the model, potentially making the training process more intricate.
>
> Thank you for highlighting this important point. In fact by allowing these parameters to be learned from the data, we address limitations associated with separate hyperparameter tuning processes, and therefore simplifying the training process. Moreover, this also enables the use of vectorized forgetting coefficients instead of scalars and as a result it increases the model’s expressivity. Unlike scalar coefficients, which impose uniform weights on all features, vectorized coefficients assign individual weights to each feature. Moreover, the ablation study in Section 4.5 and in Appendix A.2 demonstrate that these enhancements lead to improved performance. We have added a paragraph to discuss this in the training process in Section 3.2 on page 6, and reproduced below:
>
> *Compared to the GS approach, allowing the key parameters ($\vec{\alpha}$, $\vec{\beta}$, and $W$) to be learned from the data overcomes the limitations of separate tuning processes, thereby simplifying the training procedure. Additionally, this enables the use of vectorized forgetting coefficients instead of scalar ones, which significantly enhances the model's expressivity.*
>
> > Limitations of Mini-Batch Inference: Although mini-batch inference accelerates the inference speed, it may lack the flexibility of single-sample inference in certain scenarios.
>
> Thank you for raising this point. Our method is fully online and supports both single-sample and mini-batch inference. We retain the flexibility to use either approach depending on the specific scenario. Mini-batch inference, when appropriate, improves both training and inference speed, providing efficiency without sacrificing the ability to use single-sample inference when needed. In our experiments, we employed mini-batch inference to compare with other experimental setups used in state-of-the-art papers, to align with standard practices. We clarified that our method can be easily used online on page 7, and reproduced here:
>
> *It is important to note that the DGS method is fully online and supports both single-sample and mini-batch inference, offering flexibility depending on the scenario. Mini-batch inference, when suitable, improves both training and inference speeds even further. In our experiments (Section 4), we utilize mini-batch inference to ensure comparability with other experimental setups used in state-of-the-art studies.*
>
>
> > Dataset Dependence: Experimental results show that DGS performs less well on some datasets, such as the Mooc dataset, possibly due to overfitting or underfitting issues related to the specific characteristics of the dataset.
>
> We agree with this point. Performance variations across datasets, including the Mooc dataset, may be attributed to specific dataset characteristics like class imbalance. We also see similar performance fluctuations for baseline methods. Despite this, our method generally shows superior performance compared to state-of-the-art techniques across most datasets and tasks.
>
>
> > Additional empirical studies are recommended, particularly since the current performance is not promising.
>
> Thank you for your valuable feedback. In addition to the extensive experiments outlined in the paper, we conducted further investigations, including ablation studies on forward-mode automatic differentiation and exploring a different normalization method in place of softmax. These new experiments have been included in Appendix A.4 and A.5.
>
> Regarding our method’s performance, as you noted in an earlier comment, while there are some limitations on the Mooc dataset, our Deep Graph Sprints (DGS) method consistently outperforms state-of-the-art techniques across other datasets and tasks. DGS ranks first in both transductive and inductive link prediction and achieves second place in node classification by a small margin. Besides performance, it is important to note the other objectives of DGS: DGS remains significantly faster than existing methods, and addresses the limitations of GS as we now discuss on page 5 and in the new table 4.

---

### Review · Reviewer_QxQC · 2024-08-29

**Summary Of Contributions:**

The paper introduces a technique Deep-Graph-Sprints (DGS) for real-time representation learning of Continuous Time Dynamic Graphs (CTDGs), optimizing latency and enhancing the ability to capture long-term dependencies.

**Audience:**

Yes

**Claims And Evidence:**

No

**Requested Changes:**

- Add more comparison of GS and DGS techniques. Does GS have both ER and NN components? What are the differences in each of them?
- Can you compare the memory required for each approach in the evaluation?
- Add ablation study to show improvement due to the proposed forward-mode AD in comparison to the other methods for AD
- Compare DGS against additional low-latency graph learning approaches like ETSMLP (Chu & Lin (2024))
- In section 4, can you include more discussion on how AUC is calculated in the evaluation

**Strengths And Weaknesses:**

Strengths:
1. The paper presents a novel technique for dynamic graph learning.
2. The evaluation of the paper shows meaningful improvements in node classification and link prediction tasks, in terms of accuracy and latency.

Weakness:
1. The paper extensively talks about the memory requirements of all the prior works. However, no evaluation compares the actual memory use during the experiments.
2. One of the main issues for me is that - the similarities and differences between DGS and GS need to be more clearly highlighted. Is equation 1 & 2 the only difference between these techniques?
3. section 3.2 proposes forward-mode AD for the differentiation process in the ER component. However, there is no ablation study showing its usefulness.
4. contribution 3 “up to 12x faster inference on our benchmarks” is a comparison to the weakest benchmark in the evaluation. The comparisons in the contribution should be against the SOTA baselines.

> The results indicate that DGS achieves competitive performance while improving inference speed up to 12x compared to other deep learning approaches on our tested benchmarks.

Instead of comparing with the weakest baseline in the evaluation, can you state the comparison with the state-of-the-art?

> Additionally, we explain our method during inference, demonstrating how it achieves low latencies.

I don’t understand what this line means

---

> ### Author Response · Authors · 2024-09-18
> **Response to Reviewer QxQc #1**
>
> We thank reviewer QxQc for the valuable feedback. Below we replicate the identified weaknesses/requested changes and how we addressed these concerns:
>
> > Can you compare the memory required for each approach in the evaluation?
>
> Thank you for raising this point. We recognize that the current writing may be unclear regarding the memory concerns, which can be divided into three key issues:
>
> * **Memory Requirements for Temporal Graphs in Backpropagation:** Training neural networks on temporal graphs using backpropagation demands significant memory (proportional to the amount of nodes in the causal history of the event to be processed). To alleviate this, truncated backpropagation through time (TBPTT) is often used. While TBPTT reduces memory usage, it does so at the cost of capturing long-term dependencies.
> * **Applying Real-time Recurrent Learning (RTRL):** RTRL has fixed memory requirements per event instead of  being proportional to the event history but typically introduces excessive computational overhead. In Section 2, page 2, we provide a detailed comparison between RTRL and backpropagation, focusing on their computational complexities. Backpropagation operates with a complexity of O(l × d²), making it more efficient than RTRL, which has a complexity of O(d4).
>
> * **Neighbor Lookups:** A significant memory-related challenge arises from the lookup operations required for sampling neighbors in methods that depend on this process, such as RandomWalk or traditional GNN techniques. When the graph size exceeds memory capacity, these methods are forced to rely on expensive disk reads for neighbor sampling.
>
>
> Therefore, RTRL overcomes the memory limitations of backpropagation when dealing with dynamic graphs, but at the cost of unfeasible computational complexity. Our method addresses this issue by designing an architecture for which RTRL has the same complexity as backpropagation. Moreover, since our method only requires the previous states of the two nodes sharing the current edge and no sampling needs to be done this improves the efficiency further.
>
>
> To clarify these points we have added a new paragraph in Section 2 on page 2, and clarified sections 3, 5, and 6, reproduced below:
>
>
> **Changes to Section 2**
>
> *In summary, while full backpropagation through time  has high memory demands, TBPTT presents a compromise by reducing memory and computational needs at the expense of long-term dependency capture. Forward mode AD (RTRL in our case) addresses both long-term retention and memory efficiency but is restricted by its computational complexity.*
>
> *To address these limitations, our approach leverages RTRL, thus benefiting from its low memory footprint, and combines it with a custom architecture that reduces its computational complexity to match that of backpropagation. This design effectively mitigates the computational challenges typically associated with forward-mode AD, and allows the model to capture long-term dependencies unlike TBPTT.*
>
>
> **Changes to Section 3**
>
> * *This strategy not only aids in reducing computational and memory demands by limiting the dependency of each state element to a specific segment of the embedding matrix and thereby lowering the Jacobian's dimensionality but also introduces a modular structure.*
> * *Furthermore, we detail the training paradigm that distinguishes our method, highlighting its memory demands and ability to capture long-term dependencies compared to existing approaches through its RTRL-based approach.*
>
> **Changes to Section 5**
>
> *GNNs are powerful for representation learning of graphs, but adapting them to extensive datasets poses challenges. When the graph does not fit in memory, sampling the neighbors of a node may necessitate costly disk reads.*
>
> **Changes to Section 6**
>
> *CTDGs are essential for representing connected and evolving systems. The computational and memory demands associated with performing lookups to sample multiple neighbors limit their feasibility in low-latency scenarios.*

---

> > ### Author Response · Authors · 2024-09-18
> > **Response to Reviewer QxQc #2**
> >
> > > Add more comparison of GS and DGS techniques. Does GS have both ER and NN components? What are the differences in each of them?
> >
> > Thank you for your feedback.Indeed, while equations (1) and (2) highlight a key difference between GS and DGS, there are several additional aspects that distinguish the two methods. GS is a feature engineering approach with fixed embeddings, while DGS employs deep learning with learnable embeddings. Moreover, parameter optimization in GS is separate, whereas DGS allows end-to-end optimization. To address your concern, we have enhanced Section 3.1 page 5, by adding the paragraph replicated below, and also summarized the differences in Table 4 on page 15.
> >
> > *Although Equation 1 (GS) and Equation 2 (DGS) present similarities. There are several aspects that distinguish Graph-Sprints and DGS methods. GS is a feature engineering approach with fixed embeddings, while DGS employs deep learning with learnable embeddings. Moreover, parameter optimization in GS is separate, whereas DGS allows end-to-end optimization. Table 1 compares both methods.*
> >
> > > Add ablation study to show improvement due to the proposed forward-mode AD in comparison to the other methods for AD
> >
> > We thank you for your feedback and recognize the importance of demonstrating the effectiveness of the proposed forward-mode AD. To address this, we conducted an ablation study that specifically evaluates the performance improvements achieved through the use of forward-mode AD compared to truncated backpropagation as used in SOTA methods such as TGN (we remind the reviewer that full backpropagation is not feasible due to memory limitations, detailed in our earlier comments).
> >
> > This study is included in Appendix A.4, specifically on page 16, table 7. As can be seen, using RTRL instead of backpropagation is beneficial, reaching better performance throughout. We discussed these results in a new section on page 16, reproduced here:
> >
> >
> > *A.4 Ablation Study*
> >
> > *We evaluate the performance of different variants of the DGS method on the node classification task. We compare three distinct approaches:*
> > * *DGS-bp: DGS with the typical truncated backpropagation strategy as used in SOTA methods such as TGN and JODIE.*
> > * *DGS-sum: DGS using a divide-by-sum normalization technique, which simplifies the normalization process by summing activations across the network.*
> > * *DGS (proposed): Our proposed method employs forward-mode AD and utilizes softmax normalization.*
> >
> > *As shown in Table 7, our proposed DGS method outperforms both DGS-bp and DGS-sum, demonstrating the advantage of using forward-mode AD and softmax normalization for this task.*
> >
> >
> > > contribution 3 “up to 12x faster inference on our benchmarks” is a comparison to the weakest benchmark in the evaluation. The comparisons in the contribution should be against the SOTA baselines.
> >
> > Thank you for highlighting this issue. Here it depends whether we consider SOTA on performance (where TGN-attn is generally considered the best) vs. SOTA on latency (where methods like TGN-ID are better). We agree that the story is therefore more nuanced, and the important result is that our method typically provides very good performance while being the fastests. We updated the statement accordingly to the above, as reproduced below:
> >
> > **Abstract:** *... The results indicate that DGS achieves competitive performance while inference speed improves between 4x and 12x compared to other deep learning approaches on our benchmark datasets.*
> >
> > **Section 1:** *...while achieving significantly faster inference speeds—up to 12x faster than TGN-attn, and up to 8x faster than both TGN-ID and Jodie,...*
> >
> >
> > > Compare DGS against additional low-latency graph learning approaches like ETSMLP (Chu & Lin (2024))
> >
> > Thank you for the suggestion. We clarified in Section 5.2 that ETSMLP is specifically designed for sequence modeling, not for graph-based tasks. As ETSMLP is not directly comparable to DGS in the context of graph learning, we focused our comparisons on low-latency graph learning approaches that are more relevant. We updated the text to reflect this distinction, as reproduced below:
> >
> > *Chu & Lin (2024) propose ETSMLP, a model that leverages an exponential smoothing technique to model long-term dependencies in sequence learning. However, ETSMLP is tailored specifically for sequence modeling and is not applicable  to graph-based tasks out-of-the-box.*

---

> > > ### Author Response · Authors · 2024-09-18
> > > **Response to Reviewer QxQc #3**
> > >
> > > > In section 4, can you include more discussion on how AUC is calculated in the evaluation
> > >
> > > Thank you for pointing this out. For the calculation of AUC for node classification tasks dealing with binary classification and class imbalance, we calculate area under the ROC curve (ROC-AUC) by plotting the True Positive Rate (TPR) against the False Positive Rate (FPR) across various classification thresholds and computing the area under the curve. The AUC is computed using the 'sklearn' library. We clarified this in Section 4.2, reproduced below:
> > >
> > >
> > > *4.2 Node classification*
> > > *In the node classification task, given the dataset characteristics detailed in Table 5, we address binary classification with class imbalance. To evaluate performance, we calculate the area under the ROC curve (ROC-AUC), referred to as AUC for brevity, by plotting the True Positive Rate (TPR) against the False Positive Rate (FPR) across various classification thresholds, and then computing the area under the curve. The AUC is calculated using the ’sklearn’ library in python.*

---

### Review · Reviewer_Zxjc · 2024-09-04

**Summary Of Contributions:**

The authors implement a graph learning approach to maintain accuracy but increase speed of inference, leveraging components of prior graph learning approaches. The authors implement both a forward and backwards mode autodifferentiation to learn model parameters.

**Audience:**

Yes

**Claims And Evidence:**

Yes

**Requested Changes:**

Can the authors describe the significance of histogram bins for the Graph Sprint method? This seems significantly different than the embedding approach employed in the method described here.

“The notation ... denotes the concatenation of the results obtained by applying the m softmax functions…. This strategy not only aids in reducing computational and memory demands by lowering the Jacobian’s dimensionality….” Do you have any evidence to substantiate this claim? Why not take any other norm? Is it significantly worse to leave it unconstrained? If you cut out the softmax function, would it be even faster?

Would it be train the entire model using just backwards differentiation? It’s unclear to me why this component is necessary, and only seems to make this approach more difficult to more generally apply. Is it because of reduced memory overhead because of sequence length?

For inference comparison, which baselines occur on a CPU and GPU? Is it possible to implement all on a GPU, or perform them all on the CPU too?

Similar to an LSTM, can the forgetting coefficients be based of the feature input? This seems more flexible than being static across the graph.

“...but also introduces a modular structure akin to multi-head attention mechanisms in transformers (Vaswani et al., 2017), this technique has the potential to learn different information per softmax (further details in Section 3.2). “ I don’t believe this statement is accurate. In Vaswani 2017, the different attention heads performs m LxL operations across the entire sequence, where each token can be thought of as a “node” in the sequence graph. This expressivity is not in your implementation.

Do you use stochastic gradient descent to optimize any parameters? If so, which ones, and how? If so, how does SGD or its variants work for forward vs reverse mode differentiation?

**Strengths And Weaknesses:**

Strengths

Good formulation of the problem of accuracy vs speed–I appreciate that you are up front about the problem you want to solve.
Discussion of graph modeling is also great. I understand the discrete vs continuous components.

I think the ablation study of the different components is helpful. It is good to see that predictive performance is approximately the same, but inference is much faster.

The discussion of forward vs reverse mode differentiation was helpful.

The figures are very clear and helpful when following the paper. It was very easy to understand which components were optimized in each particular manner.

The transductive vs inductive split is an interesting comparison across methods.

Weaknesses

My main concern is there are a number of complex components in which ablations are not performed, but are still discussed in this work. In particular, the forward-mode differentiation is novel and interesting, but it is not clear whether this is actually necessary. In particular, since you had to code up this forward diff manually, this seems like a hinderance for others trying to apply this method.

While the discussion of forward vs reverse differentiation was helpful, it would be nice to discuss how that affected the design decisions you made with your model, particularly the computational complexity. Moreover, your method employs a neural network with many more parameters and operations that is optimized via backwards autodiff. What percent of compute do you really save by doing forward autodiff with \alpha , \beta and W? Seems very minimal.

It is still unclear to me how the softmax function “lower[s] the Jacobian’s dimensionality…”, especially if you just didn’t include the softmax at all.

---

> ### Author Response · Authors · 2024-09-18
> **Response to Reviewer Zxjc #1**
>
> We thank reviewer Zxjc for the valuable feedback. Below we replicate the requested Changes and how we addressed these concerns:
>
> > Can the authors describe the significance of histogram bins for the Graph Sprint method? This seems significantly different than the embedding approach employed in the method described here.
>
> To clarify and avoid confusion we rephrased "histogram bins" to "bucketed features", where numerical features are represented using a one-hot vector based on predetermined buckets (e.g. bucket edges denoting the percentiles). The bucketed features provide a deterministic encoding of the input feature values, needing manual tuning to determine the buckets, and results in sparse and large feature vectors. This differs significantly from the DGS embedding approach that leverages a learnable embedding matrix aiming to address the weaknesses of the GS approach. Specifically, the mapping to the embeddings is learned from the data instead of being fixed, avoiding the manual tuning phase Moreover, the resulting embedding vectors are dense and consequently can be chosen to be much smaller.
> We have updated the writing in Section 3.1.1 adding this information and reproduced below:
>
>
> Here, $\vec{S}\_t$ denotes the state of a node at time $t$, evolving from its previous state $\vec{S}\_{t-1}$ and the state of its immediate neighbor $\vec{S}^*\_{t-1}$, while also incorporating the current edge's features $F\_t$, encoded through the $\vec{\delta}$ function, which bucketizes these features. The $\vec{\delta}$ function maps numerical features into a one-hot encoded vector based on predetermined buckets (e.g., buckets corresponding to specific percentiles). This bucketization provides a deterministic representation of the input features, which requires manual tuning to define the buckets and results in sparse, high-dimensional feature vectors. The coefficients $\alpha$ and $\beta$ are scalar forgetting factors that modulate the impact of neighborhood and past information on the current state.
>
>
>
> > “The notation ... denotes the concatenation of the results obtained by applying the m softmax functions…. This strategy not only aids in reducing computational and memory demands by lowering the Jacobian’s dimensionality….” Do you have any evidence to substantiate this claim? Why not take any other norm? Is it significantly worse to leave it unconstrained? If you cut out the softmax function, would it be even faster?
>
> Thank you for your question. Removing the softmax function would indeed simplify gradient calculations but at the cost of losing the nonlinearity essential for expressive learning. In fact, removing a non-linearity would result in a purely linear algorithm, which therefore could at most learn linear decision boundaries. We experimented with alternative norms, including one that divides by the sum, and included these in our ablation study in appendix (A.4) in table 7 on page 16. As indicated in the results table, using this simpler norm resulted in worse performance, underscoring the benefit of the softmax function. While exploring other norms or even unconstrained activation functions could be a promising direction for future work, our current experiments suggest that the softmax provides a significant advantage. Regarding inference speed, the divide-by-sum approach showed similar performance to the softmax, as detailed in the comparative speed table provided below and added to appendix (A.5) in table 8. We reproduced the added sections below (tables can be seen in the revised pdf):
>
>
>
> **A.4 Ablation Study**
>
> We evaluate the performance of different variants of the DGS method on the node classification task. We compare three distinct approaches:
> * DGS-bp: DGS with truncated backpropagation, which limits the backpropagation depth to reduce computational complexity.
> * DGS-sum: DGS using a divide-by-sum normalization technique, which simplifies the normalization process by summing activations across the network.
> * DGS (proposed): Our proposed method employs forward-mode AD and utilizes softmax normalization.
>
> As shown in Table 7, our proposed DGS method outperforms both DGS-bp and DGS-sum, demonstrating the advantage of using forward-mode AD and softmax normalization for this task.
>
> **A.5 Comparison of Inference Speed**
>
> We evaluate and compare the inference speed of two different methods: DGS (proposed) and DGS-sum. The comparison is based on their performance across various datasets. Table 8 summarizes the average inference times, in seconds, along with their standard deviations. The results indicate that both methods exhibit similar inference speeds across the different datasets.

---

> > ### Author Response · Authors · 2024-09-18
> > **Response to Reviewer Zxjc #2**
> >
> > > Would it be train the entire model using just backwards differentiation? It’s unclear to me why this component is necessary, and only seems to make this approach more difficult to more generally apply. Is it because of reduced memory overhead because of sequence length?
> >
> > Thank you for your question. Indeed, as the reviewer pointed out, the memory overhead would become too large due to the whole graph history preceding an event would be needed in the batch. In other words, training the entire model using full backward differentiation is not feasible due to memory constraints, as it would require storing the entire computational graph in gpu memory, which is impractical for large graphs. This is a typical problem in graphs, and why all SOTA methods have some form of truncation. **We have added a discussion on this problem regarding dynamic graphs on page 2, and reproduced here**:
> >
> > *Temporal models, such as recurrent neural networks (RNNs) and GNNs for temporal graphs, pose specific challenges for backpropagation due to their memory-intensive requirements. The memory complexity for storing intermediate states across a history significantly impacts the feasibility of full backpropagation. For instance, in an RNN with sequence length $l$ and state size $d$, backpropagation-through-time exhibits computational and memory complexities of $\mathcal{O}(l \times d^2)$, posing scalability issues for long sequences~\citep{baydin2018automatic}.*
> >
> >
> > Therefore, current SOTA methods use truncated backpropagation through time (BPTT), but at the cost of losing the ability to learn long-term dependencies. Our method leverages forward-mode automatic differentiation, which maintains the same computational complexity as backward-mode (due to our architecture choices). This approach enables the learning of long-term dependencies that truncated BPTT cannot capture, at the cost of a less expressive architecture. **We provide an additional ablation study that includes results from using truncated BPTT in the DGS method (table 7 on page 16)**, demonstrating its inferior performance and underscoring the importance of forward-mode automatic differentiation for our approach. Below we reproduce the relevant part of the added table:
> >
> >
> >
> > *We evaluate the performance of different variants of the DGS method on the node classification task. We compare three distinct approaches:*
> > * *DGS-bp: DGS with the typical truncated backpropagation strategy as used in SOTA methods such as TGN and JODIE.*
> > * *DGS (proposed): Our proposed method employs forward-mode AD.*
> >
> > *As shown in Table 7, our proposed DGS method outperforms both DGS-bp, demonstrating the advantage of using forward-mode AD for this task.*
> >
> >
> > > For inference comparison, which baselines occur on a CPU and GPU? Is it possible to implement all on a GPU, or perform them all on the CPU too?
> >
> > In our experiments, all baselines were executed on the same GPU (NVIDIA GeForce RTX 2080 Ti GPU with 11GB of memory) to ensure a fair comparison. However, it is indeed possible to run all the baselines on a CPU as well. We believe that this should not affect the conclusions, and chose to use GPU since it is the most common processor to run neural networks on. We clarified this information is mentioned in section 4.4, page 9.
> >
> > *… Tests were performed on a Linux PC equipped with 24 Intel Xeon CPU cores (3.70GHz) and an NVIDIA GeForce RTX 2080 Ti GPU (11GB). Note that all experiments, including those for link prediction and node classification mentioned in the previous sections, used the same machine.*
> >
> >
> > > Similar to an LSTM, can the forgetting coefficients be based of the feature input? This seems more flexible than being static across the graph.
> >
> > Thank you for raising the point. Indeed, adapting the parameters alpha and beta based on input data is a promising direction for future work. This enhancement aims to significantly increase the model's adaptability to varying input features, thereby improving its overall performance. The approach involves the introduction of trainable matrices 𝑊1 and 𝑊2, which dynamically adjust alpha and beta in response to specific input characteristics. To ensure that alpha and beta stay within the range of 0 to 1, a sigmoid function can be applied as in gated recurrent neural networks. We have added a paragraph in the discussion, and reproduce it here:
> >
> > *Future work includes exploring alternative normalization functions or activation functions beyond softmax and incorporating advanced optimization algorithms such as Adam to replace the current use of stochastic gradient descent for updating DGS parameters during forward-mode AD. Additionally, a significant enhancement under consideration is enabling input-dependent adaptability for the parameters $\vec{\alpha}$ and $\vec{\beta}$, aiming to improve the model's responsiveness to varying input features and enhance overall performance, similar to approaches used in gated recurrent neural networks.*

---

> > > ### Author Response · Authors · 2024-09-18
> > > **Response to Reviewer Zxjc #3**
> > >
> > > > “...but also introduces a modular structure akin to multi-head attention mechanisms in transformers (Vaswani et al., 2017), this technique has the potential to learn different information per softmax (further details in Section 3.2). “ I don’t believe this statement is accurate. In Vaswani 2017, the different attention heads performs m LxL operations across the entire sequence, where each token can be thought of as a “node” in the sequence graph. This expressivity is not in your implementation.
> > >
> > >
> > > We appreciate your critique and acknowledge that the analogy may have lacked clarity. The comparison we intended to make is that, like multi-head attention, our model benefits from a modular design that allows it to learn various representations from the inputs, albeit with a different methodology. We agree that this phrasing may confuse the reader, and the similarity to multiple heads is superficial. We removed this comparison accordingly.
> > >
> > > > Do you use stochastic gradient descent to optimize any parameters? If so, which ones, and how? If so, how does SGD or its variants work for forward vs reverse mode differentiation?
> > >
> > > Thank you for the question. We indeed update all parameters (alpha, beta, W) by SGD. The key difference between forward vs reverse mode is how we compute the gradients, but once computed, the gradients are exactly the same for both methods (this is a mathematical guarantee, not part of our proposal). Then, we apply SGD as usual, by updating the parameters taking a little step in the direction of the negative gradient.
> > >
> > > Regarding the computation of gradients using forward-mode vs backprop, we discuss the differences in methodology in section 2. All gradients computed using backprop make use of the built-in functionality of pyTorch, while we wrote custom methods in pyTorch to compute gradients using forward-mode.

---

### Author Response · Authors · 2024-09-18

We thank the reviewers for their thorough review and constructive feedback. We believe we have addressed all comments and with those, the manuscript has become stronger. We have submitted a revised manuscript and also added the specific answers to the reviewers comments below.

---

### Author Response · Authors · 2024-11-07

Thank you to the Reviewers and Action Editor for their careful evaluation and constructive feedback, which have strengthened our work. The final camera-ready version, along with a link to a video presentation, has been submitted.

Sincerely,
The Authors

---

### Decision · Action_Editor_rzGY · 2024-10-11

**Recommendation:** Accept as is

**Comment:**

Reviewers were unanimous in recommending that the paper be accepted.  There were some critiques around the significance and breadth of experimentation.  The authors addressed some of the critiques during the rebuttal period.

**Audience:**

The contribution is of interest to some of TMLR readership.

**Claims And Evidence:**

Claims are generally well supported by evidence, though there is room for improvement in terms of the breadth of experimentation.